# Spatial Transcriptomic Analysis Reveals Regional Transcript Changes in Early and Late Stages of rd1 Model Mice with Retinitis Pigmentosa

**DOI:** 10.3390/ijms241914869

**Published:** 2023-10-03

**Authors:** Ying Zhou, Yuqi Sheng, Min Pan, Jing Tu, Xiangwei Zhao, Qinyu Ge, Zuhong Lu

**Affiliations:** 1State Key Laboratory of Bioelectronics, School of Biological Science & Medical Engineering, Southeast University, Nanjing 210096, China; yingzhou@seu.edu.cn (Y.Z.); sheng_yuqi@163.com (Y.S.); jtu@seu.edu.cn (J.T.); xwzhao@seu.edu.cn (X.Z.); zhlu@seu.edu.cn (Z.L.); 2School of Medicine, Southeast University, Nanjing 210097, China; panmin1120@163.com

**Keywords:** spatial transcriptomic, retinitis pigmentosa, degenerative disease, apoptosis of photoreceptor cells, neovascularization

## Abstract

Retinitis pigmentosa (RP) is the leading cause of inherited blindness with a genetically heterogeneous disorder. Currently, there is no effective treatment that can protect vision for those with RP. In recent decades, the rd1 mouse has been used to study the pathological mechanisms of RP. Molecular biological studies using rd1 mice have clarified the mechanism of the apoptosis of photoreceptor cells in the early stage of RP. However, the pathological changes in RP over time remain unclear. The unknown pathology mechanism of RP over time and the difficulty of clinical treatment make it urgent to perform more refined and spatially informed molecular biology studies of RP. In this study, spatial transcriptomic analysis is used to study the changes in different retinal layers of rd1 mice at different ages. The results demonstrate the pattern of photoreceptor apoptosis between rd1 mice and the control group. Not only was oxidative stress enhanced in the late stage of RP, but it was accompanied by an up-regulation of the VEGF pathway. Analysis of temporal kinetic trends has further identified patterns of changes in the key pathways of the early and late stages, to help understand the important pathogenesis of RP. Overall, the application of spatial transcriptomics to rd1 mice can help to elucidate the important pathogenesis of RP involving photoreceptor apoptosis and retinal remodeling.

## 1. Introduction

Retinitis pigmentosa (RP) is a group of retinal degenerative diseases characterized by progressive apoptosis of the photoreceptor cells and injuries in the retinal pigment epithelium (RPE) cell layer [1,2]. To date, rd1 model mice have been commonly used for more than 30 years to study the pathogenesis of RP [3,4]. The retinal degeneration in rd1 mice is remarkably rapid, with the peak of rod death at around P14 [5]. Several mechanisms are involved in the pathogenesis of rd1 retinas, including the apoptosis of retinal photoreceptor cells and the structure remodeling of the retina [6,7]. There is an ever-growing amount of research on the therapeutic protocols for RP [8,9]. It is crucial to identify the pathogenesis of the different retinal layers and the stages of RP in order to develop gene therapy. The main objective of treating RP is to salvage the function of retinal photoreceptors and the functional architecture of the RPE cell layer. However, there is still a lack of in-depth molecular biology studies of the rd1 retina aimed at gaining a deeper understanding of the pathogenesis of RP and addressing the unavailability of effective treatments for RP.

Currently, research on the pathogenesis of RP primarily focuses on its genetic heterogeneity and photoreceptor cell damage [10,11]. The pathogenesis of RP in rd1 mice has been investigated using a variety of techniques, including transcriptomics [12], Western Blot-based proteomics [13,14] and microarray-based fluorescent quantitative PCR [15]. Previous studies have been able to elucidate the changes in gene expression associated with photoreceptor apoptosis in the early stages of RP and it has been suggested that RP is closely related to angiogenic pathways and neurodegenerative diseases. However, further validation of the relevant changes in the angiogenic pathway in RP, as well as a more detailed elucidation of the pathogenesis of RP, remains to be investigated. Previous studies have found that RP can affect different retinal layers, such as photoreceptor cell apoptosis mainly located in the outer nuclear layer, while changes in other layers are not yet known [16,17,18]. The study of the differences in the retinal layer is crucial for elucidating the pathogenesis of RP. There may be significant differences between the different retinal layers that have varying degrees of impact on the progression of RP. It requires the use of techniques that preserve spatial location information to further identify the differences between different retinal layers in RP. In this study, spatial transcriptomic technology was used instead of single-cell RNA-seq. Spatial transcriptomic technology is able to investigate spatial location and the level of gene expression simultaneously [19]. It can provide new insights into the spatial and temporal heterogeneity to understand the biomolecular mechanisms associated with RP. To date, spatial transcriptomic technology has not been used to study the effects of different stages on different layers of the rd1 mouse retina.

In this study, the Smart-seq2 transcriptomic library building method and laser capture microscopy (LCM) were combined to obtain samples from different retinal layers of rd1 mice at different ages [20]. LCM enables the careful dissection of different cells from tissues that have been snap frozen [21,22]. LCM has been coupled with RNA extraction methods to analyze the transcriptome of distinct tissues using RNA-seq [23,24]. This sampling technique, which enables precise separation of different retinal layers, is the key point of this study. Target tissue samples were obtained by sampling each layer of frozen retinal slices using LCM. Afterwards, regional transcript changes were analyzed between different retinal layers. This protocol not only allows the robust acquisition of samples from different retinal layers while retaining spatial information, but also allows the analysis of differentially expressed genes (DEGs) at different ages. The results showed that photoreceptor apoptosis was present in both the early and late stages of RP. Furthermore, not only was oxidative stress enhanced in the late stage of RP, but it was accompanied by an up-regulation of the VEGF pathway. Ultimately, it can help to develop effective therapies by elucidating the pathogenesis of the early and late stages of RP at different spatial locations.

## 2. Results

### 2.1. Spatial Transcriptomic Analysis in rd1 Mice

In this study, spatial transcriptomic analysis was performed on frozen sections of eyes from C57BL/6J mice (*n* = 4) and rd1 model mice (*n* = 4) with RP (Figure 1A). The age range of the mice selected for this study was from P14 to P35. The application of spatial transcriptomic technology in rd1 mice allowed for efficient sampling and differential expression analysis of the different retinal layers between the RP and control groups. The mean number of genes obtained per spatial spot was above 10,000 for both control mice and rd1 mice, and the difference between the two was not statistically significant (Figure 1B). Three replicate samples each of outer retina, middle retina and inner retina were obtained from 100 μm adjacent to the optic nerve in rd1 mice and control mice; see Appendix A for the specific sampling strategy. The specific information of the samples is provided in Appendix A. A total of 72 samples were obtained for sequencing analysis, with an average sequencing depth of 20 M and an average gene count of 13,756. Comparison of sagittal sections of eyeballs from rd1 and C57BL/6J mice at different ages showed that rd1 model mice (Appendix A) had thinner total retinal thickness (Appendix A) and outer nuclear layer (ONL) thickness (Appendix A) compared to control mice (Appendix A) (*p* < 0.01). The difference in retinal thickness between the two groups was statistically significant and not limited by age (*p* < 0.01).

### 2.2. The Time Course Is Divided into Early and Late Periods

The rd1 mice demonstrated rapid photoreceptor apoptosis compared to control mice, with the ONL layer becoming thinner and thinner over time. Few photoreceptor cells were left at P28 for the rd1 mice. According to the time course of the apoptosis of photoreceptor cells, P14 and P21 were defined as early stages while P28 and P35 were defined as late stages of RP, using the K-means hierarchical clustering method (Appendix A). It was possible to distinguish the rd1 and control groups by PCA after dividing early and late stages (Figure 1C). The time progression of the transition from early to late stages of RP can be seen through the PCA clustering method (Figure 1D). The PCA clustering results for the early and late stages of the different layers are shown in Appendix A. The heat maps of early stage RP (Figure 2A) and late stage RP (Figure 2B) indicated that it is possible to distinguish between up-regulated and down-regulated genes and pathways in the early and late stages. The DEGs of the early and late stages of RP are shown in Appendix A. The analysis of DEGs between the RP and control groups was performed using separate volcano maps for the early (Figure 2C) and late stages (Figure 2D). The trends of co-expressed genes between the RP and control groups differed from early to late stages. The co-expressed genes of the control group suggested an overall decreasing trend over time in the different sampling strata (Figure 2E). The co-expressed genes in the RP group suggested an overall trend of up-regulation over time in the different sampling strata (Figure 2F). The DEGs in the early and late stages of the different sampling strata are shown in Figure 2G.

### 2.3. Identification of DEGs in the Early and Late Stages of RP

There was down-regulation of genes related to the phototransduction pathway, such as Pde6b, Rho and Gnat1, in both early and late stages of RP. The down-regulation of genes related to the phototransduction pathway may be due to the absence of photoreceptor cells. However, both rod cell loss and a reduced expression of phototransduction-related genes are supposed to characterize RP. There was up-regulation of genes related to oxidative stress and vasoproliferative activity, such as mt-Nd1, mt-Nd2, mt-Atp6, Efnb2, Kdr and Nr2f2, in the late stage of RP. A list of DEGs for each spatial stratum at each time point is shown in Appendix A. The DEGs were defined by adjusted *p*-values ≤ 0.05 and Log_2_|Fold change| > 1. A comparison of DEGs from different sampling strata at different times is shown in Appendix A. Among them, genes involved in the activation, inactivation, recovery and regulation of the phototransduction cascade, such as Rcvrn, Rgs9 and Pde6g, showed down-regulation both in the early and late stages. They are also involved in the regulation of the G protein-coupled receptor signaling pathway in RP. In addition, genes associated with phototransduction, the typical retinal recycling of optic rods (twilight vision) pathway and the apoptosis of photoreceptor cells such as Pde6b, Nrl and Gnat1, showed down-regulation in rd1 mice at different sampling levels and at different time points (Figure 3). The relevant marker genes involved in the neovascular pathway, such as Efnb2, Kdr and Nr2f2, showed up-regulation in rd1 mice at different sampling levels and at different time points (Figure 3). The heat map is based on Log_2_Fold change, with red suggesting enhancement and green suggesting reduction.

### 2.4. Validation by Quantitative PCR

To further validate the results of the spatiotemporal transcriptomic study, the levels of selected transcripts in the different sampling strata of the rd1 mice and control groups at four time points were determined using real-time quantitative PCR. The specific quantitative PCR primer sequences used are shown in Appendix A. The differences in gene expression between the rd1 mice and control groups were compared as relative expression at different times (P14, P21, P28 and P35) and in different sampling strata (IRL, MRL and ORL). Transcripts associated with phototransduction (Rho, Pde6b, Gnat1 and Nrl) appeared to be down-regulated in the different layers of the rd1 group (Appendix A) and at different times (Appendix A). The results of quantitative PCR validation were consistent with the results of spatial transcriptomic experiments. Photoreceptor apoptosis was prevalent during the development of the early and late stages of RP. In addition, angiogenic pathway-related genes, such as Efnb2, Kdr and Nr2f2, were up-regulated relative to the control groups in all three sampling strata (Appendix A) and at all four time points (Appendix A). Statistical differences were found between the different time groups and different retinal layer groups by the statistical method of ANOVA (*p* < 0.05).

### 2.5. Immunofluorescence of VEGF Pathway in RP

The Efnb2 gene in the vascular endothelial growth factor (VEGF) pathway not only acts synergistically with the Src gene, but also promotes endothelial cell migration. Significantly, Efnb2 activates the MAPK signaling pathway and promotes the proliferation of endothelial cells. The KEGG simulations demonstrating their close connection are shown in Appendix A. There is a close relationship between photoreceptor apoptosis (Rho, Nrl and Crx) and the VEGF pathway (Nr2f2, Efnb2 and VEGFA) in the development of RP. The network of protein interactions between them is shown in Appendix A. VEGF plays an important role in RP, while Efnb2 has a potentially important role in the VEGF pathway. Immunofluorescence on slide samples from different periods for Efnb2 was performed to investigate the possible localization of neovascularization in the process of RP. In the control mice, immunoreactivity for Efnb2 was located throughout the retinal layer (Appendix A). The results showed that the expression of Efnb2 in control mice was generally unchanged between ages (Appendix A). The expression of Efnb2 was significantly enhanced in rd1 mice relative to the control group and was higher near the outer retinal layer (Figure 4).

### 2.6. GO Pathway Trends over Time in the Early and Late Stages of RP

The comparison of GO pathway results between early and late stages of RP revealed that the GO pathway in the early stage of RP focused on visual perception, phototransduction, photoreceptor cell differentiation and photoreceptor cell maintenance (Figure 5A). The GO pathway in the late stage of RP is focused on phototransduction, photoreceptor cell differentiation, mitochondrial electron transport, NADH to ubiquinone and cell morphogenesis involved in differentiation (Figure 5B). The detailed results of the comparison of GO pathways in the early and late stages of RP are shown in Appendix A. Both the control group (Figure 5C) and the RP group (Figure 5D) showed down-regulation over time of genes associated with photoreceptor apoptosis, such as Rho, Gnat1, Pde6b, Pde6a, Crx and Rcvrn. Genes involved in oxidative stress, such as Ndufa4, Cox7b, Atp6v1a, Ndufb8, Ndufv1 and Gucy2f, showed a down-regulation trend over time in the control group (Figure 6A), but an up-regulation trend over time in the RP group (Figure 6B). Genes involved in neovascularization, such as Efnb2, Kdr, Nr2f2, Tfap2b, Prox1 and Slc8a1, showed a down-regulation trend over time in the control group (Figure 6C), but an up-regulation trend over time in the RP group (Figure 6D). The DEGs between the early and late stages of RP are also involved in the development of neurodegenerative disease. Ultimately, the temporal dynamics of the expression profiles of genes associated with neurodegenerative diseases, such as Ndufa2, Ndufb6, Snca, Ndufb8 and Cox8a, were characterized by trending (Appendix A). Furthermore, it is possible to identify potential time series patterns in expression profiles to help understand the dynamic patterns of genes and how they are functionally linked.

### 2.7. Fitting of Early and Late Stages at Different Spatial Spots

The normalized expression of key pathway-related genes in the RP group was fitted to the control group. Figure 7 shows the results of the fitting between the RP and control groups for each spatial sampling point. The heat map demonstrates the difference in the coefficient of fitting. In both the early stage (Figure 7A) and the late stage (Figure 7B), the coefficient of fitting for each spatial spot of the phototransduction pathway showed a downward trend. There is an upward trend of genes related to the neurodegenerative pathway in the inner retinal layer at the early stage (Figure 7C). In other retinal layers of the early stage and in all the retinal layers of the late stage, there is a downward trend of the neurodegenerative pathway in the rd1 group compared to the control group (Figure 7D). The closer the square of R in the linear regression equation is to 1, the closer the fitted curve is to the actual curve. The 95% confidence intervals are added with dashed lines. The fitting results and the formulas of R-squared and RMSE for each group are shown in Appendix A. The BISQUARE method was used to perform a robust regression process that can effectively deal with discrete values of gene expression. The fitting results showed that the coefficient of fitting was significantly lower in the late stage than in the early stage of RP. In addition, the genes related to the neurodegenerative pathway clearly showed different fitting results from those of the phototransduction pathway. The coefficient of fitting was significantly higher in the IRL in the early stage, while the coefficients of fitting in all other strata and in the late stage were reduced in the RP group compared to the control group. The preservation of spatial location information allows for better analysis of the developmental patterns of RP in different retinal layers.

## 3. Materials and Methods

### 3.1. Animals

There were two groups of mice, including the rd1 mice group with retinitis pigmentosa (rd1, Pde6bem1Cin purchased from Gem Pharmatech Co., Ltd., Nanjing, China) and the control mice group (C57BL/6J, purchased from Gem Pharmatech Co., Ltd.). Other variables were controlled for both groups, such as equal numbers of males and females. The animals used in this study were all males. All samples were collected between 1:00 pm and 2:00 pm, to prevent potential effects from circadian rhythms and the onset of light. The mice were be euthanized by cervical subluxation after anesthesia.

### 3.2. Eye Acquisition and Sectioning

After euthanizing the mice, the eyeballs were removed and then placed in a PBS buffer to be washed twice. The eyeballs were then embedded with OCT (Bioss Antibodies, Woburn, MA, USA) and placed in liquid nitrogen for rapid freezing. Frozen sections were then cut to 15 μm thickness using a microtome (Thermo Fisher Scientific, Waltham, MA, USA, Thermo Scientific Cryotome FSE 230V). Sections of each mouse were visualized and laser microdissected using a microscope (ZEISS, Oberkochen, Germany, ZEISS microscope, PALM MicroBeam). The sections containing the optic nerve were selected from each group to determine sampling locations (Appendix A). Three replicates were taken by laser microdissection from the outer retinal layer (ORL), middle retinal layer (MRL) and inner retinal layer (IRL) near the optic nerve. Each sampling site was spaced 100 μm apart, and the diameter of the sampling sites was approximately 40 μm. A total of 72 samples were sampled from rd1 mice and control C57BL/6J mice at four time points, including P14, P21, P28 and P35. There was also a study that chose to study rd1 mice starting from P14 [15]. The spatial location of the sampled loci can be precisely preserved.

### 3.3. Sampling and Library Construction

Three replicate samples of the ORL, MRL and IRL were obtained from 100 μm adjacent to the optic nerve in the rd1 mice and the control mice. The results of the actual sampled tissue sections, after HE staining, are shown in Appendix A. Some of the choroidal and RPE layers overlapped during sectioning. Library construction was performed on samples according to the Smart-seq2 library construction protocol. Tissues sampled at the single cell level were lysed and then transformed into templates for first-strand cDNA synthesis using TSO oligonucleotides. The cDNA was then pre-amplified by PCR using KAPA HiFi enzyme and IS PCR primers. The amplified cDNA was adsorbed and purified by 1:1 Ampure XP magnetic beads. Then, 1 ng of cDNA template was interrupted with transposase Tn5. Afterwards, 17 cycles of enrichment PCR were performed with P5 and P7 junctions at both ends. Finally, subsequent sequencing was performed.

### 3.4. Quantitative PCR

The cycle threshold (Ct value) is the number of cycles experienced by the fluorescent signal in each reaction when it reaches a set threshold. The samples were quantified by an internal reference method. The internal reference was the glyceraldehyde-3-phosphate dehydrogenase (GAPDH) of the mouse. The PCR amplification system consisted of the DNA amplification enzyme TB Green Premix Ex Taq II (Takara, RR420A, 2×), upstream and downstream primers (10 μm), a DNA template and enzyme-free water to make up the volume to 20 μL. The primer sequences are shown in Appendix A. The formula is as follows:(1)$${\Delta Ct}_{disease/control}={Ct}_{marker\;gene}-{Ct}_{internal\;reference}$$
(2)$$\Delta \Delta Ct={\Delta Ct}_{disease}-{\Delta Ct}_{control}$$

### 3.5. Immunofluorescence of Frozen Sections

First, the frozen sections were placed in an oven at 37 °C for 10 min to control the moisture. The frozen sections were then fixed in 4% paraformaldehyde for 30 min and washed three times in PBS (pH 7.4) on a decolorization shaker for 5 min each time to fix the frozen sections. The tissue sections were placed in a repair cassette filled with EDTA antigen repair buffer (pH 8.0) for antigen repair in a microwave oven. After natural cooling, the slides were washed three times for 5 min each time in PBS (pH 7.4). The sections were shaken slightly on a decolorization shaker. Then, the sections were circled around the tissue with a histochemical pen to prevent antibody runoff. The PBS was then shaken off. BSA was added drop wise, and the sections were closed for 30 min. Then, 10% donkey serum was used for the primary antibody of goat origin and 3% BSA for the primary antibody of another origin. The blocking solution was gently shaken off. A drop of PBS was added with a certain proportion of the primary antibody on the section. The manufacturer of the antibody used and its dilution is shown in Appendix A. The section was incubated flat in a wet box at 4 °C overnight. The slides were then washed three times in PBS (pH 7.4) on a decolorization shaker for 5 min each time. Autofluorescence quencher was added to the circle for 5 min and rinsed under running water for 10 min. The sections were shaken dry and sealed with an anti-fluorescence quenching sealer. Sections were observed under a fluorescent microscope (LEICA, Wetzlar, Germany, DM2500) and images were collected (DAPI UV excitation wavelength is 330–380 nm, emission wavelength is 420 nm, blue light; CY3 excitation wavelength is 510–560 nm, emission wavelength is 590 nm, red light).

### 3.6. Data Analysis

Raw data for each locus were obtained by Illumina HiSeq 2500 sequencing. Clean data were obtained by quality control. Comparisons were made by Hisat, and raw counts were obtained by Featurecounts and normalized by fragments per kilo-base per million mapped reads (FPKM). GO enrichment analysis of DEGs was performed using the Metascape website [25]. Corrected *p*-values < 0.01 were considered to indicate significantly enriched GO terms. The DEGs between the disease and control groups were obtained using DEseq2 and defined by adjusted FDR < 0.05, *p*-values ≤ 0.05 and Log_2_|Fold changes| > 1. The results of the quantitative PCR were calculated by ANOVA. The differences between the groups were statistically significant at *p*-values < 0.05. In addition, linear regression, a regression method that models the relationship between the independent and dependent variables linearly, was performed using the POLYNOMINAL function in MALAB (R2018a). The linear regression equation is:(3)y=ax+b
where a is the fitting coefficient and b is a constant. The fitting coefficient with the smallest overall error is obtained by the least squares method, which is calculated as:(4)$$a=\frac{\sum_{1}^{n}\begin{array}{cc} \left(x_{i}-\bar{x}\right) & \left(y_{i}-\bar{y}\right) \end{array}}{\sum_{1}^{n}{\left(x_{i}-\bar{x}\right)}^{2}}$$

Moreover, the closer the square of R is to 1, the closer the fitted curve is to the actual curve. The 95% confidence intervals are indicated by dashed lines. The BISQUARE method was used to perform a robust regression process, which can effectively deal with the discrete values of the gene expression. The R-squared and RMSE for each group are shown in Appendix A.

## 4. Discussion

There have been many studies using rd1 mice to investigate the pathogenesis of RP. The main goal of the studies is to help diagnose and treat patients with RP. The early stage of RP is characterized by a severe loss of photoreceptors, which leads to an imbalance between oxygen supply and consumption. Afterwards, it leads to vascular regression, inflammation and gliosis. There is an increase in the stress response in the late stage of RP. An overall retinal remodeling affects neurons and glial cells with increased inflammation and oxidative responses [26]. The differences in pathological changes in the early and late stages make it extremely difficult to treat patients with RP at present. The late stage of RP is characterized by a complete loss of vision and possible neovascularization following the apoptosis of photoreceptor cells, further making the treatment difficult. An improved understanding of the differences in the pathological mechanisms of the early and late stages of RP is important to elucidate the disease progression. In this study, the disease progression in different retinal layers of rd1 mice at early and late stages was analyzed using spatial transcriptomics. The use of spatial transcriptomics allows a deeper understanding of the complex structure of transcriptional networks, spatially dependent biological processes and the heterogeneity of retinal diseases. The results demonstrate photoreceptor apoptosis in both early and late stages of RP. Genes up-regulated in proximal samples are mainly involved in neuronal apoptosis, positive regulation of synaptic transmission and ATP metabolic process. Genes up-regulated in distal samples are mainly involved in visual perception and visual phototransduction (Appendix A). Furthermore, there is a significant difference between P14 and P21, before the retinal structure was significantly disorganized. The DEGs between P14 and P21 are mainly involved in the ATP metabolic process and energy derivation by the oxidation of organic compounds. GO analysis shows the down-regulation of P21 in lens development in camera-type eye and sensory organ development (Appendix A). Not only is oxidative stress enhanced in the late stage of RP, but it is accompanied by an up-regulation of the VEGF pathway. Analysis of temporal kinetic trends has further identified patterns of changes in the key pathways of the early and late stages to help understand the important pathogenesis of RP.

The present study not only reveals important pathological changes in the late stages of RP, but also suggests that neovascularization may occur in conjunction with a structural remodeling of the retina. It has important implications for understanding the physiological and pathological vascular growth in RP. The available studies have not yet been able to elucidate the mechanisms of the main pathological changes in the late stages of RP. Moreover, more and more cases of RP with choroidal neovascularization (CNV) are being reported [27,28]. These patients can be effectively treated with intravitreal injections of anti-VEGF. Neovascularization may be an important cause of the poor prognosis of RP due to the enhanced oxidative stress and increased reactive oxygen species (ROS) in the late stage of RP [29,30]. It has been suggested that severe posterior and peripheral vasculitis of RP patients may as a result of the presence of neovascularization. The mechanism of the RP with CNV has not been elucidated. Enhanced oxidative stress and changes in the VEGF pathway may be the main causes of alterations in the choroidal vascular system in RP. The Eph receptor ligand and transmembrane protein ephrin B2 (Efnb2) is a key gene in the VEGF pathway and a key transcript in vascular remodeling in advanced RP. It has been elucidated that VEGF expression is unchanged in young rd1 mice using Western Blot analysis [14]. However, it was confirmed in the present study that there is a difference in the expression of EFNB2 between the rd1 mice and the control group using immunofluorescence. The present study not only demonstrates the synergistic relationship between Efnb2 and VEGF, but the results of immunofluorescence also showed that Efnb2 is most highly expressed in the retinal location close to the RPE layer. VEGFA and EFNB2 have been shown to be closely related to neovascularization [31,32]. The present study further validates that both VEGFA and EFNB2 may be involved in neovascularization in rd1 mice.

The ways in which Efnb2 affects neovascularization and its involvement in the VEGF pathway have been explained for a long time. The activation of Efnb2 signaling induces changes in cell proliferation and migration to regulate retinal angiogenesis [33]. The increased expression of Efnb2 is closely associated with inflammation and apoptosis [34]. It is an important regulator of VEGF receptor endocytosis and downstream signaling [35]. Two distinct pathways have been shown to regulate Efnb2: one is stimulated by Ca^2+^ influx, and the other is adrenergic B2 ligands, which regulate the proteolytic processing of Efnb2 receptors and their complexes [36]. Stimulation of Efnb2 in retinal endothelial cells has been shown to produce significant migration, which is regulated by induction of the MAPK pathway following Akt phosphorylation [37,38]. Stimulation of Efnb2 can lead to significant migration and proliferation of retinal endothelial cells. Previous studies have highlighted Efnb2 as an important target for anti-angiogenesis [39]. Blocking Efnb2 may be an attractive alternative or combination anti-angiogenic therapeutic strategy to disrupt the role of VEGF in the pathological angiogenesis of RP. A deficiency of Efnb2 can effectively inhibit neoangiogenesis. Injections by the subretinal and suprachoroidal routes are preferred over the intravitreal route to treat patients with RP and CNV. The spatially resolved transcriptomics used in this study can effectively contribute to the development of targeted therapies for RP.

The DEGs in the late stage of RP, when compared to the early stage, were found to be involved not only in the VEGF pathway, but also in the pathway of neurodegenerative diseases. This is consistent with the studies that have found retinal molecules to be associated with degenerative diseases such as AD and PD [40]. At present, retinas are expected to be a potential future diagnostic tool for neurodegenerative diseases. In vitro models of the retina may be useful for the early diagnosis of age-related diseases such as AD and PD [41]. The present study revealed differences in the development of neurodegenerative diseases between the RP and control groups. Moreover, the important causative genes for age-related diseases, including RP and neurodegenerative disease, showed a pronounced down-regulation trend in the late stage than the early stage. It was found that the photoreceptor apoptotic pathway was more significantly decreased in RP. Furthermore, there was a tendency for the neurodegenerative pathway to be slightly up-regulated in the IRL in the early stages, which may be linked to the connection of ganglion cell layers to the brain. Both retinal degenerative diseases and neurodegenerative diseases require not only further molecular biological studies to elucidate their important pathogenesis, but also important studies that can elucidate the link between them. This is the direction of our future application of spatial transcriptomic analysis. 

This study was mainly limited by the sample size (*n* = 4) and sampling level of rd1 mice used for spatial transcriptomic analysis. Moreover, spatial transcriptomic analysis is not single-cell and contains about ten cells per sampling spot, which does not distinguish well between the effects of cell types of RP. Additionally, Smart-seq2 only captures mRNA. Finally, although this study initially confirmed an association between RP and neurodegenerative diseases, the underlying mechanisms of their development are not clear and further studies of the association, using spatial transcriptomics, are needed.

## 5. Conclusions

In this study, spatial transcriptomic analysis with a precise sampling method was used to analyze the progression of RP in different retinal layers at early and late stages. The results demonstrate the pattern of photoreceptor apoptosis between rd1 and control groups. Not only was oxidative stress enhanced in the late stage of RP, but it was accompanied by an up-regulation of the VEGF pathway. Furthermore, the analysis of the time course has further identified patterns of the changes in the key pathways in the early and late stages to help understand the important pathogenesis of RP. This study further advances not only the current knowledge of pathological changes in the early and late stages of RP, but also the precise diagnosis and targeting of therapies.

## Figures and Tables

**Figure 1 ijms-24-14869-f001:**
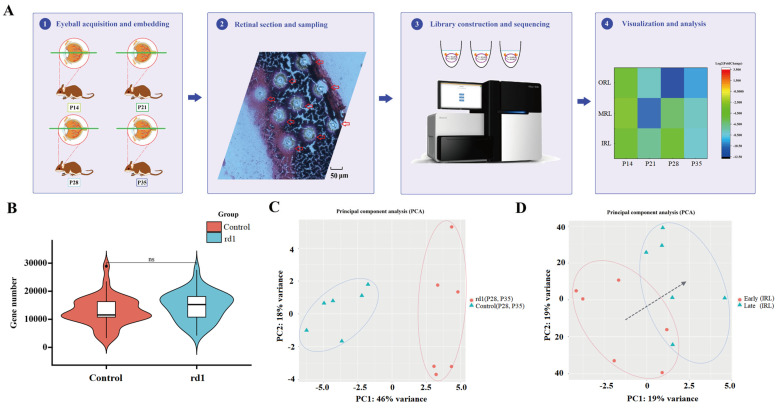
Application of spatial transcriptomics in retinitis pigmentosa. (**A**) The workflow of the application of spatial transcriptomic analysis in retinitis pigmentosa (RP). The rd1 model mice and control C57BL/6J mice of the same age were used in this study. In general, the process is divided into four steps. The first step is the collection and embedding of the eyeball. The second step is the sectioning and sampling of the retina. The third step is the construction of libraries and sequencing of samples. The fourth step is the analysis and visualization of spatial transcriptomic data. (**B**) The comparison of the number of genes obtained from the rd1 and control mice. There was no significant difference between the number of genes obtained in the disease and control groups. (**C**) The different time-stratified clusters were grouped into early and late stages according to the K-means method. In the late stages (P28, P35), it was possible to separate the rd1 and control mice using a principal component analysis (PCA) cluster map. (**D**) PCA clustering maps of the early and late stages of RP showed a clear trend from early to late stages. (The black arrow represents the trend.)

**Figure 2 ijms-24-14869-f002:**
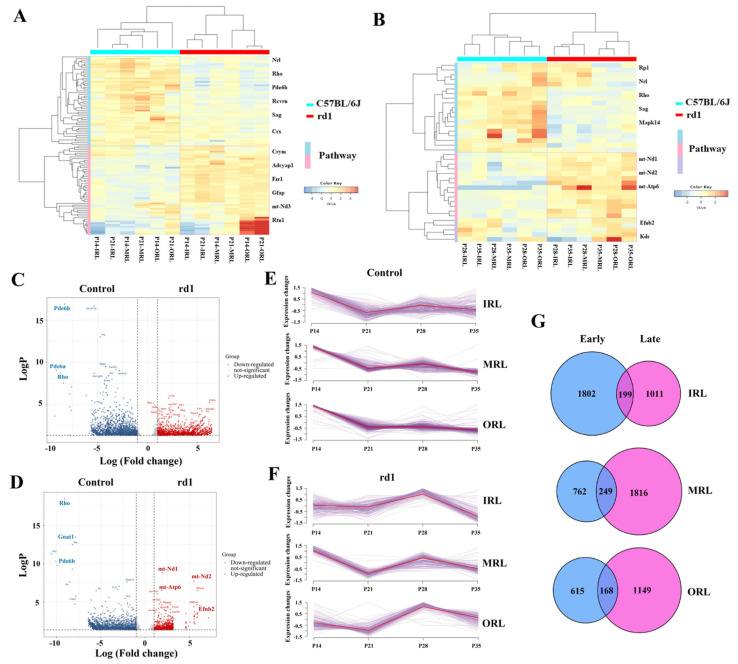
Differences between early and late stages of retinitis pigmentosa. (**A**) The heat map of the early stage of retinitis pigmentosa (RP) showed reduced expression of the phototransduction-related genes, such as Pde6b, Rho and Nrl, compared to the control group. P14 and P21 are early stages and P28 and P35 are late stages. Outer retinal layer is abbreviated as ORL. Middle retinal layer is abbreviated as MRL. Inner retinal layer is abbreviated as IRL. (**B**) The heat map of the late stage of RP showed decreased expression of genes involved in phototransduction such as Rho, Pde6b and Nrl. Additionally, the expression of genes involved in NADH dehydrogenase (ubiquinone) activity and oxidative phosphorylation pathway, such as mt-Nd1, mt-Nd2 and mt-Atp6, were elevated. Moreover, there is an up-regulation of genes related to pathways involved in cell migration for sprouting angiogenesis, such as Efnb2 and Kdr. (**C**) Volcano map of the early stage of RP showed the down-regulation of the genes of Rho, Pde6b and Pde6a, and the up-regulation of the genes mt-Nd3, Gfap and Grifin. (**D**) Volcano map of the late stage of RP showed the down-regulation of the genes of Rho, Pde6b, and Gnat1 and the up-regulation of the genes mt-Nd1, mt-Nd2 and Efnb2. (**E**) The number of differentially expressed genes was compared in the early and late stages of the different sampling strata. (**F**) The expression of genes showed down-regulation trends at four time points of the data in the control group. (**G**) The expression of genes indicated different trends from the control group at four time points of the data in the RP group.

**Figure 3 ijms-24-14869-f003:**
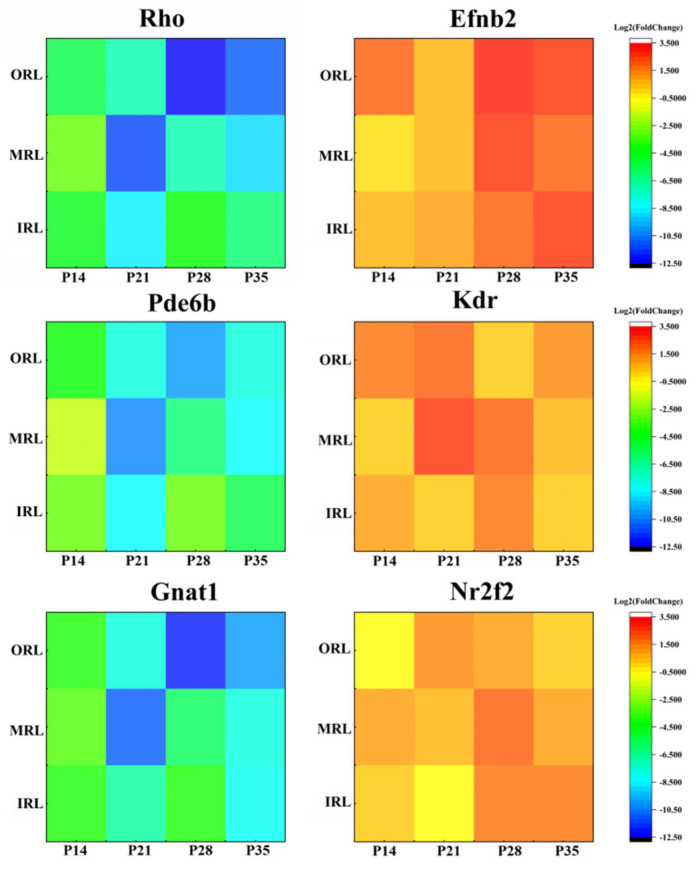
The heat map of the expression levels of the genes involved in photoreceptor apoptosis (Rho, Pde6b and Gnat1) showed down-regulation in all the retinal layers and different times. Meanwhile, the heat map of the expression levels of the genes involved in the vascular endothelial growth factor pathway (Efnb2, Kdr and Nr2f2) showed up-regulation in all the retinal layers and different times. The value of Log_2_Fold change > 0 means up-regulation and the value < 0 means down-regulation, when comparing the rd1 mouse group to the control mouse group.

**Figure 4 ijms-24-14869-f004:**
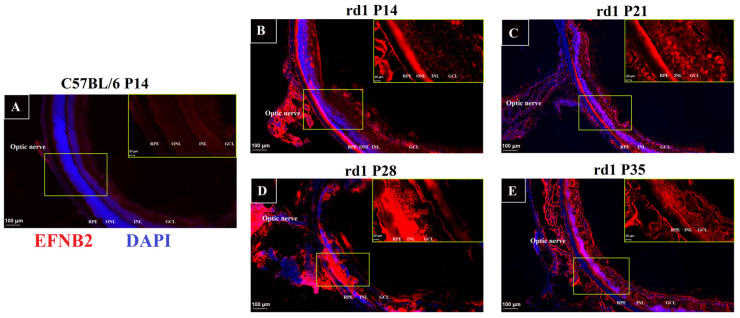
Immunofluorescence validation of the vascular endothelial growth factor (VEGF) pathway. (**A**) In control mice, immunoreactivity of EFNB2 was located throughout the retinal layers and remained generally unchanged across ages. The scale bars in the figures are 100 μm. The scale bars in the enlarged image are 20 μm. (See Appendix A for other time points). (**B**) The expression results of EFNB2 in rd1 mice at day P14. (**C**) The expression results of EFNB2 in rd1 mice at day P21. (**D**) The expression results of Efnb2 in rd1 mice at day P28. (**E**) The expression results of Efnb2 in rd1 mice at day P35.

**Figure 5 ijms-24-14869-f005:**
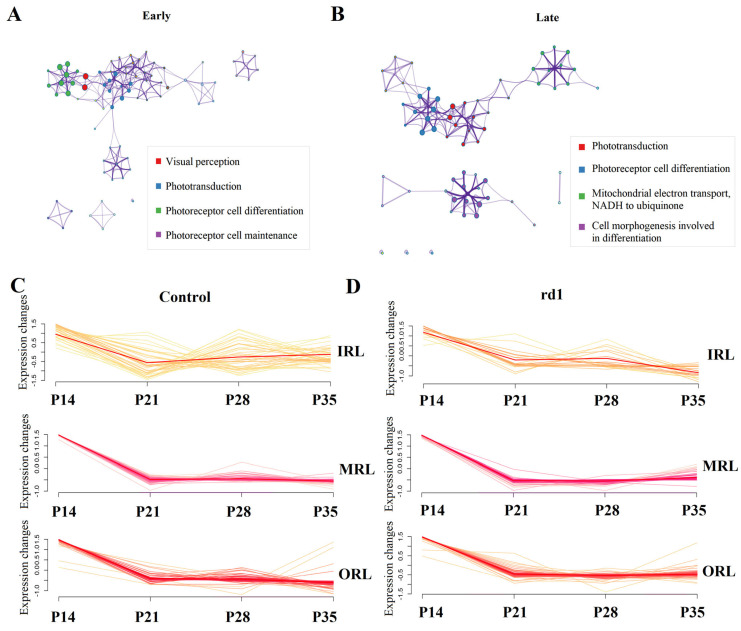
Comparison of the GO pathways in the early and late stages of retinitis pigmentosa. (**A**) The GO pathways in early stages of RP were focused on visual perception, phototransduction, photoreceptor cell differentiation and photoreceptor cell maintenance. (**B**) The GO pathways in late stages of RP were focused on phototransduction, photoreceptor cell differentiation, mitochondrial electron transport, NADH to ubiquinone and cell morphogenesis involved in differentiation. (**C**) There is a downward trend of the expression of photoreceptor cells apoptosis-related genes, such as Rho, Gnat1, Pde6b, Pde6a, Crx and Rcvrn, in the control group at different retinal layers and times. (**D**) There is a downward trend of the expression of photoreceptor cells apoptosis-related genes, such as Rho, Gnat1, Pde6b, Pde6a, Crx and Rcvrn, in the RP group at different retinal layers and times.

**Figure 6 ijms-24-14869-f006:**
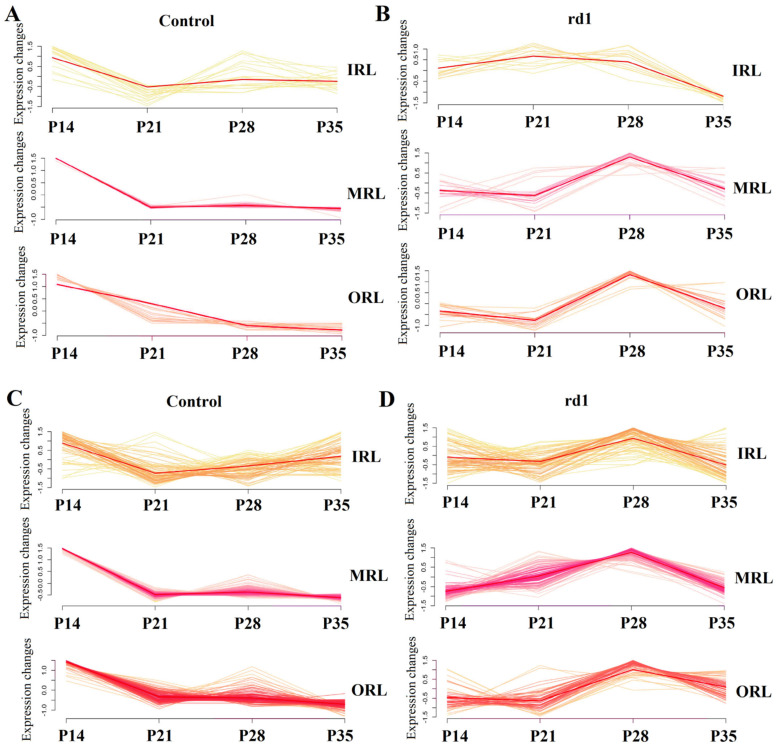
Different trends of the expression of the genes between the rd1 and control groups. There is a downward trend (**A**) and an upward trend (**B**) of the expression of the genes related to oxidative stress, including Ndufa4, Cox7b, Atp6v1a, Ndufb8, Ndufv1 and Gucy2f, in the control group at different retinal layers and times. There is a downward trend (**C**) and an upward trend (**D**) of the expression of the genes related to neovascularization, including Efnb2, Kdr, Nr2f2, Tfap2b, Prox1 and Slc8a1, in the control group at different retinal layers and times.

**Figure 7 ijms-24-14869-f007:**
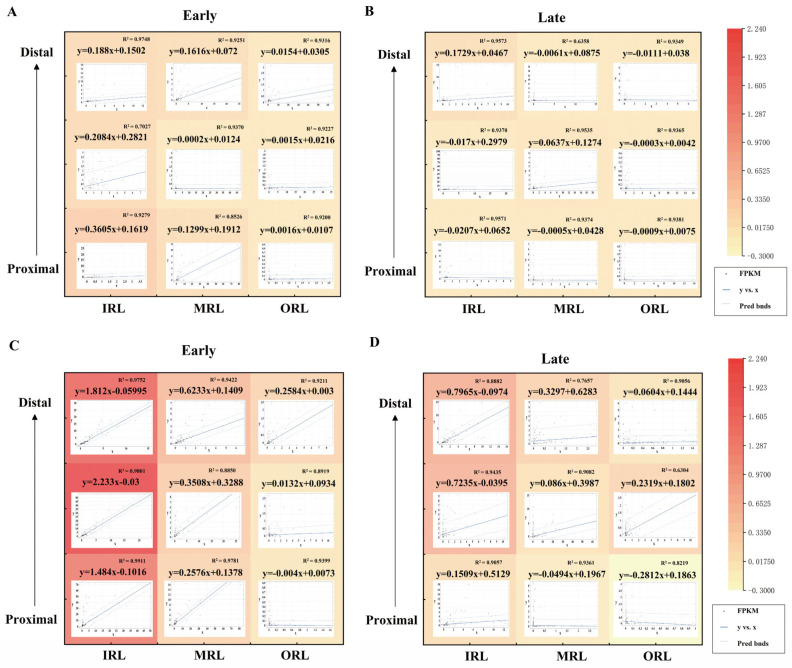
The fitting between the rd1 and control groups at the early and late stages. In both the early (**A**) and late stage (**B**), the coefficient of fitting for each spatial spot of the phototransduction pathway showed a downward trend. The fitting coefficients are all less than 1. The heat map shows the difference in the coefficient of fitting. (**C**)There is an upward trend of the genes related to the neurodegenerative pathway in the inner retinal layer at the early stage. The fitting coefficients for the inner retinal layer of the early stage were greater than 1. In other retinal layers of the early stage and in all the retinal layers of the late stage (**D**), there is a downward trend of the neurodegenerative pathway in the rd1 group compared to the control group.

## Data Availability

The data used in this paper can be downloaded from the NCBI website: https://www.ncbi.nlm.nih.gov/bioproject/?term=PRJNA826720 accessed on 14 April 2022.

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
