# Peer review of "Spatial Transcriptomic Analysis Reveals Regional Transcript Changes in Early and Late Stages of rd1 Model Mice with Retinitis Pigmentosa"

_ijms, 2023, doi:10.3390/ijms241914869_

Round 1
Reviewer 1 Report
Review of IJMS Manuscript ijms-2569614 “Spatial transcriptomic reveals the changes of early and late stages of rd1 model mice with retinitis pigmentosa.”
In this manuscript, the authors seek to identify changes in the spatial transcriptome in different retinal layers of rd1 mice, a common mouse model used to uncover the pathogenesis of retinal degeneration (RD). The purpose is to uncover a lack of in-depth molecular biology studies of the rd1 retina, although why this is necessary is not fully understood; there have been many studies over three decades that reveal the pathogenic mechanisms of RD in this model. While the writing is challenging to read in parts, the data reveal that apoptosis occurred, and oxidative stress and VEGF upregulation increased in later stages of RD. Altogether the data are not surprising and novel but do reinforce what is known in the field. The following items should be addressed prior to resubmission:
How is spatial transcriptomic analysis performed: on the tissue itself? How? PCR was described, and immunohistochemistry as well, but not how transcripts were identified spatially. Were transcripts isolated in retinal layers and compared among layers? If so, I would consider using the phrase, “regional transcript changes” instead of spatial transcriptomics, as that implies transcript changes within a small region using the latest technology. This is confusing.
Sex of animals used in this study needs to be addressed.
Line 61-62 “The spatial transcriptomic information of different retinal layers in the early and late stages of RP can be analyzed comprehensively.”
The introduction does not offer a testable hypothesis, only a methodology applied to a well-studied mouse model. The introduction does not clarify why different retinal layers are of interest or worthy of dissecting apart.
Line 76 “…collected between 12:00 am to 2:00 pm to prevent potential effects from the circadian…”
Possibly a typo/mix-up but as written 12:00 am (midnight) to 2:00 pm is not a well-controlled time-window with respect to light onset.
Line 79 “After executing the mice…”
Consider “euthanize” rather than “execute”. Also specify the method of euthanasia.
Line 83 “…observed with a microscope (ZEISS microscope).”
Please specify the model of microscope used.
Line 85 “…replicates were taken by laser micro-dissection…”
Please provide a prior citation that describes the micro-dissection in more detail.
Line 126 “…proportion of the primary antibody on the section.”
Please specify the antibodies used, the antibody manufacturer, and dilutions. This can be added as a supplemental table.
Line 130-131 “Sections were observed under a fluorescent microscope…”
Please specify the model of microscope used.
Line 135 “…Illumina HiSaq 2500…”
Typo: HiSeq
Line 159-160 “The age range of the mice selected for this study was from P14 to P35.”
The age range of the mice requires additional consideration. From published literature and the authors Supplemental Figure 2, the retina and outer segments are significantly diminished at the first time-point, and by the end of the time-course, the rods are completely absent. Presumably the authors are detecting transcription changes related to cell-death pathways and cell loss rather than what leads to rod degeneration due to the rd1 mutation.
Line 227 “There was down-regulation of genes related to the phototransduction pathway such as Pde6b, Rho and Gnat1 in both early and late stages of RP.”
The authors do not consider here nor in the discussion that down-regulation is attributable to photoreceptor loss. This needs to be considered and addressed.
Line 242 “…heat map is based on Log2Foldchange…”
Typo: Log2Fold change
Figure 4 and 5
qPCR of select genes that were also quantified in Smart-seq2 is repetitive. Such validation may be better used in Supplemental Figures. Consider replacing in the manuscript with IHC of VEGF (from supplemental) and other vasculature components since that appears to be one of the major focuses and findings of the paper.
Figure 8
The figure does not appear to be a sufficient resolution and is difficult to read and interpret.
Line 398 “The present study is not limited to the retina, but also includes the RPE layer.”
The RPE is not referenced in any of the results or sampling technique.
Line 478-480 “Moreover, the spatial transcriptomics is not single-cell and contains about ten cells per sampling spot, which does not distinguish well between the effects of cell types of RP.”
Authors should also acknowledge that Smart-seq(2) only captures mRNA.
An English editor should be used to avoid using adjectives and adverbs as nouns throughout the document, and it should be edited for ease of reading.
Author Response
Dear Editor and Reviewers:
First of all, thank you for your previous comments concerning our manuscript entitled “Spatial transcriptomic reveals the changes of early and late stages of rd1 model mice with retinitis pigmentosa”. Those comments are all valuable and very helpful for revising and improving our paper, as well as the important guiding significance to our research. We have carefully studied these comments and corrected the paper in the hope that it will be more suitable for International Journal of Molecular Sciences.
Responds to the Reviewers’ comments:
Reviewer 1:
In this manuscript, the authors seek to identify changes in the spatial transcriptome in different retinal layers of rd1 mice, a common mouse model used to uncover the pathogenesis of retinal degeneration (RD). The purpose is to uncover a lack of in-depth molecular biology studies of the rd1 retina, although why this is necessary is not fully understood; there have been many studies over three decades that reveal the pathogenic mechanisms of RD in this model. While the writing is challenging to read in parts, the data reveal that apoptosis occurred, and oxidative stress and VEGF upregulation increased in later stages of RD. Altogether the data are not surprising and novel but do reinforce what is known in the field. The following items should be addressed prior to resubmission:
Response:
Thank you very much for your suggestion. Firstly, all of us appreciate your valuable feedback on this paper. It is our honor to receive your approval of the manuscript. Secondly, we apologize for the difficulty in reading caused by inaccurate writing of the article. We have carefully revised the article. We hope the revised manuscript can meet your requirements. Finally, thank you again for reading and for your corrections, followed by responses to all the comments.
Major points:
Comment 1:
How is spatial transcriptomic analysis performed: on the tissue itself? How? PCR was described, and immunohistochemistry as well, but not how transcripts were identified spatially. Were transcripts isolated in retinal layers and compared among layers? If so, I would consider using the phrase, “regional transcript changes” instead of spatial transcriptomics, as that implies transcript changes within a small region using the latest technology. This is confusing.
Response to comment 1:
Thank you very much for your suggestion. Firstly, we apologize for not accurately describing the experimental method of spatial transcriptome. We have added the following content to the experimental methods section. “Target tissue samples were obtained by sampling on each layer of frozen retinal slices using LCM. Afterwards, regional transcript changes were analyzed between different retinal layers.” We also added “regional transcript changes” to the title. The final title is “Spatial transcriptomic reveals regional transcript changes of early and late stages of rd1 model mice with retinitis pigmentosa”. Finally, thank you again for your valuable suggestion.
Comment 2:
Sex of animals used in this study needs to be addressed.
Response to comment 2:
Thank you very much for your suggestion. Firstly, we apologize for not clarifying the gender of animals. Secondly, all animals used in this study were male. And, we have included this description in the experimental methods section. “The animals used in this study were all males.” Finally, thank you again for your valuable advice.
Comment 3:
Line 61-62 “The spatial transcriptomic information of different retinal layers in the early and late stages of RP can be analyzed comprehensively.”
The introduction does not offer a testable hypothesis, only a methodology applied to a well-studied mouse model. The introduction does not clarify why different retinal layers are of interest or worthy of dissecting apart.
Response to comment 3:
Thank you very much for your suggestion. Firstly, we apologize for not clarifying in the introduction why different retinal layers are of interest and not proposing testable hypotheses. Then, we added the following description in the introduction. “Previous studies have found that RP can affect different retinal layers, such as photoreceptor cell apoptosis mainly located in the outer nuclear layer, while changes in other layers are not yet known.[16-18] The study of differences in the retinal layer is crucial for elucidating the pathogenesis of RP. There may be significant differences between the different retinal layers that have varying degrees of impact on the progression of RP. It requires the use of techniques that preserve spatial location information to further identify the differences between different retinal layers in RP.” Among them, our hypothesis is that “Different retinal layers have varying degrees of impact on the progression of RP, and there may be significant differences.” Finally, thank you again for your valuable suggestion. This suggestion makes this article more meaningful.
- 16. Hwang, Y.H.; Kim, S.; Kim, Y.Y.; Na, J.H.; Kim, H.K.; Sohn, Y.H. Optic Nerve Head, Retinal Nerve Fiber Layer, and Macular Thickness Measurements in Young Patients with Retinitis Pigmentosa. Curr Eye Res 2012, 37, 914-920, doi:10.3109/02713683.2012.688163.
- 17. Gong, Y.; Xia, H.; Zhang, A.; Chen, L.J.; Chen, H. Optical coherence tomography biomarkers of photoreceptor degeneration in retinitis pigmentosa. Int Ophthalmol 2021, 41, 3949-3959, doi:10.1007/s10792-021-01964-1.
- 18. Santos, A.; Humayun, M.S.; de Juan, E.J.; Greenburg, R.J.; Marsh, M.J.; Klock, I.B.; Milam, A.H. Preservation of the inner retina in retinitis pigmentosa. A morphometric Arch Ophthalmol 1997, 115, 511-515, doi:10.1001/archopht.1997.01100150513011.
Comment 4:
Line 76 “…collected between 12:00 am to 2:00 pm to prevent potential effects from the circadian…”
Possibly a typo/mix-up but as written 12:00 am (midnight) to 2:00 pm is not a well-controlled time-window with respect to light onset.
Response to comment 4:
Thank you very much for your suggestion. There may be confusion when writing. We have changed “12:00 am (midnight)” to “1:00 pm”. Our experiment was conducted at noon to prevent potential impacts on circadian rhythms. We have tried our best to control time consistency to prevent errors. The descriptions of “12:00 am” and “12:00 pm” can both cause misunderstandings. We have made changes to the article. Finally, thank you again for your valuable suggestion.
Comment 5:
Line 79 “After executing the mice…”
Consider “euthanize” rather than “execute”. Also specify the method of euthanasia.
Response to comment 5:
Thank you very much for your suggestion. We have changed “execution” to “euthanasia”. And the following description has been added in the method section. “The mice will be euthanized by cervical subluxation after anesthesia and the eyeball will be collected.” Although the cervical subluxation method is not advocated, it is still the fastest method of euthanasia. We hope you can approve. Finally, thank you again for your valuable suggestion.
Comment 6:
Line 83 “…observed with a microscope.”
Please specify the model of microscope used.
Response to comment 6:
Thank you very much for your suggestion. The model of the microscope is ZEISS microscope, PALM MicroBeam. Moreover, it has been labeled in the article. “Sections of each mouse were visualized and laser microdissected using a microscope.(ZEISS microscope, PALM MicroBeam).”.
Comment 7:
Line 85 “…replicates were taken by laser micro-dissection…”
Please provide a prior citation that describes the micro-dissection in more detail.
Response to comment 7:
Thank you very much for your suggestion. The previous citation of laser micro-dissection has been supplemented in the main text. The laser capture microscopy have been used in previous studies. The specific description is as follows. “Laser capture microscopy (LCM) enables careful dissection of different cells from tissues that have been snap frozen.[21,22] LCM has been coupled with RNA extraction methods to analyze the transcriptome of distinct tissues using RNA-seq [23,24]. ” Finally, thank you again for your valuable advice.
- 21. Nichterwitz, S.; Chen, G.; Aguila Benitez, J.; Yilmaz, M.; Storvall, H.; Cao, M.; Sandberg, R.; Deng, Q.; Hedlund, E. Laser capture microscopy coupled with Smart-seq2 for precise spatial transcriptomic profiling. Nat Commun 2016, 7, doi:10.1038/ncomms12139.
- 22. Frost, A.R.; Eltoum, I.E.; Siegal, G.P.; Emmert Buck, M.R.; Tangrea, M.A. Laser Microdissection. Current Protocols in Molecular Biology 2015, 112, doi:10.1002/0471142727.mb25a01s112.
- 23. Pembroke, W.G.; Babbs, A.; Davies, K.E.; Ponting, C.P.; Oliver, P.L. Temporal transcriptomics suggest that twin-peaking genes reset the clock. Elife 2015, 4, doi:10.7554/eLife.10518.
- 24. Zechel, S.; Zajac, P.; Lonnerberg, P.; Ibanez, C.F.; Linnarsson, S. Topographical transcriptome mapping of the mouse medial ganglionic eminence by spatially resolved RNA-seq. Genome Biol 2014, 15, 486, doi:10.1186/s13059-014-0486-z.
Other LCM related literature is not listed in the main text. As follows:
- Hedlund, E., Karlsson, M., Osborn, T., Ludwig, W. & Isacson, O. Global gene expression profiling of somatic motor neuron populations with different vulnerability identify molecules and pathways of degeneration and protection. Brain2010, 133, 2313–2330.
- Chung, C. Y. et al. Cell type-specific gene expression of midbrain dopaminergic neurons reveals molecules involved in their vulnerability and protection. Hum. Mol. Genet. 2005, 14, 1709–1725.
- Simunovic, F. et al. Gene expression profiling of substantia nigra dopamine neurons: further insights into Parkinson’s disease pathology. Brain 2009, 132, 1795–1809.
- Saxena, S., Cabuy, E. & Caroni, P. A role for motoneuron subtype-selective ER stress in disease manifestations of FALS mice. Nat. Neurosci. 2009, 12, 627–636.
- Murray, L. M., Beauvais, A., Gibeault, S., Courtney, N. L. & Kothary, R. Transcriptional profiling of differentially vulnerable motor neurons at pre-symptomatic stage in the Smn (2b/-) mouse model of spinal muscular atrophy. Acta Neuropathol. Commun. 2015, 3, 55.
- Lobsiger, C. S., Boillee, S. & Cleveland, D. W. Toxicity from different SOD1 mutants dysregulates the complement system and the neuronal regenerative response in ALS motor neurons. Proc. Natl Acad. Sci. USA 2007, 104, 7319–7326.
- Kadkhodaei, B. et al. Transcription factor Nurr1 maintains fiber integrity and nuclear-encoded mitochondrial gene expression in dopamine neurons. Proc. Natl Acad. Sci. USA 2013, 110, 2360–2365.
- Bandyopadhyay, U. et al. RNA-Seq profiling of spinal cord motor neurons from a presymptomatic SOD1 ALS mouse. PLoS ONE 2013, 8, e53575.
Comment 8:
Line 126 “…proportion of the primary antibody on the section.”
Please specify the antibodies used, the antibody manufacturer, and dilutions. This can be added as a supplemental table.
Response to comment 8:
Thank you very much for your suggestion. Supplementary Table S2 explains the manufacturer and dilution parameters of the antibodies used. The details are as follows.
Table S2:Manufacturer of the antibody used and its dilution
Reagents |
Factory |
Product number |
Dilution ratio |
EDTA(PH8.0)Antigen Repair Fluid |
Servicebio |
G1206 |
|
BSA |
Servicebio |
G5001 |
|
autofluorescence quencher |
Servicebio |
G1221 |
|
primary antibody: VEGFA |
Servicebio |
GB13034 |
1:100 |
primary antibody: EFNB2 |
Biotech Lab |
BTL0096N |
1:100 |
DAPI |
Servicebio |
G1012 |
1:300 |
Anti-fluorescence quenching encapsulant |
Servicebio |
G1401 |
|
We have also added corresponding descriptions in the main text. “Manufacturer of the antibody used and its dilution was shown in Table S2.” Finally, thank you again for your valuable suggestion.
Comment 9:
Line 130-131 “Sections were observed under a fluorescent microscope…”
Please specify the model of microscope used.
Response to comment 9:
Thank you very much for your suggestion. The fluorescence microscope model is LEICA, DM2500. We have made modifications to the article. Thank you again for your valuable suggestion.
Comment 10:
Line 135 “…Illumina HiSaq 2500…”
Typo: HiSeq
Response to comment 10:
Thank you very much for your suggestion. The “HiSaq” in the main text has been corrected to “HiSeq”. Thank you very much for your careful reading to help us discover spelling errors.
Comment 11:
Line 159-160 “The age range of the mice selected for this study was from P14 to P35.”
The age range of the mice requires additional consideration. From published literature and the authors Supplemental Figure 2, the retina and outer segments are significantly diminished at the first time-point, and by the end of the time-course, the rods are completely absent. Presumably the authors are detecting transcription changes related to cell-death pathways and cell loss rather than what leads to rod degeneration due to the rd1 mutation.
Response to comment 11:
Thank you very much for your suggestion. Firstly, we must explain why sampling started from P14. Our study obtained time transcriptomic results of rd1 mice aged P14 to P35. The age selection for rd1 mouse research varies. There has been study also chose to start from P14.[25] As is well known, P14 is the age at which mice open their eyes. If the mice studied are younger than P14, their eyes may not yet be open. This is also feasible for understanding the pathogenesis of RP. If it is necessary to reselect the age, all experiments may need to be redone. This is a bit difficult for us.
In addition, we must acknowledge that the mutations detected in the results are related to cell loss. However, both rod cell loss and reduced expression of phototransduction-related genes are supposed to characterize RP. This article mainly discusses the differences in transcriptome changes related to RP at different times, with a focus on the up-regulation of related genes in the VEGF pathway. Photoreceptor apoptosis is the main pathogenic mechanism of RP. Besides, the down-regulation of gene expression, whether caused by cell loss or gene mutation, cannot be concluded. Finally, thank you again for your valuable suggestion.
- Hackam, A.S.; Strom, R.; Liu, D.; Qian, J.; Wang, C.; Otteson, D.; Gunatilaka, T.; Farkas, R.H.; Chowers, I.; Kageyama, M., et al. Identification of gene expression changes associated with the progression of retinal degeneration in the rd1 mouse. Invest Ophth Vis Sci 2004, 45, 2929, doi:10.1167/iovs.03-1184.
Comment 12:
Line 227 “There was down-regulation of genes related to the phototransduction pathway such as Pde6b, Rho and Gnat1 in both early and late stages of RP.”
The authors do not consider here nor in the discussion that down-regulation is attributable to photoreceptor loss. This needs to be considered and addressed.
Response to comment 12:
Thank you very much for your suggestion. Indeed, it is possible that the down-regulation of genes involved in phototransduction pathways is due to the absence of photoreceptors. However, it is also possible that it is caused by gene mutations. Photoreceptor apoptosis is the main pathogenic mechanism of RP. Besides, the down-regulation of gene expression, whether caused by cell loss or gene mutation, cannot be concluded. Photoreceptor apoptosis is the main pathogenic mechanism of RP, and whether its down-regulation is caused by loss of photoreceptor cells or by gene mutations is not something we can conclude. However, this did not prevent us from discussing the differences in RP-associated transcriptomic changes that exist at different times and discussing the up-regulation of relevant genes in the VEGF pathway. We then added the section to the discussion as follows. “The down-regulation of genes related to the phototransduction pathway may be due to the absence of photoreceptor cells. However, both rod cell loss and reduced expression of phototransduction-related genes are supposed to characterize RP. ”
Finally, thank you again for your valuable suggestions.
Comment 13:
Line 242 “…heat map is based on Log2Foldchange…”
Typo: Log2Fold change
Response to comment 13:
Thank you very much for your suggestion. The “Log2Foldchange” in the main text has been corrected to “Log2Fold change”. Thank you very much for your careful reading to help us discover spelling errors.
Comment 14:
Figure 4 and 5
qPCR of select genes that were also quantified in Smart-seq2 is repetitive. Such validation may be better used in Supplemental Figures. Consider replacing in the manuscript with IHC of VEGF (from supplemental) and other vasculature components since that appears to be one of the major focuses and findings of the paper.
Response to comment 14:
Thank you very much for your suggestion. First, we moved the results of qPCR to the supplementary FigureS5 and FigureS6. Then, we moved the IHC map of EFNB2 from the Supplementary Figures to the main text and replaced it with the clearer original images. Finally, the changes in VEGF-related pathways were included as one of the main focus findings of this thesis. Thank you again for your valuable suggestions. The updated in-text figure is shown below.
Figure 4. Immunofluorescence validation of the Vascular endothelial growth factor (VEGF) pathway.
(A) In control mice, immunoreactivity of EFNB2 was located throughout the retinal layers and remained generally unchanged across ages. (See Figure S10 for other time points) (B) The expression results of EFNB2 in rd1 mice at day P14. (C) The expression results of EFNB2 in rd1 mice at day P21. (D) The expression results of Efnb2 in rd1 mice at day P28. (E) The expression results of Efnb2 in rd1 mice at day P35.
Comment 15:
Figure 8
The figure does not appear to be a sufficient resolution and is difficult to read and interpret.
Response to comment 15:
Thank you very much for your suggestion. We are very sorry that the resolution of original Figure 8 is not high enough. We have made modifications to the diagram. In addition, we have added image descriptions to make it easier to read and interpret. The latest image is shown below.
Figure 7. The fitting between rd1 and control group at the early and late stages.
In both the early (A) and late stage (B), the coefficient of fitting for each spatial spot of the phototransduction pathway showed a downward trend. The fitting coefficients are all less than 1. The heat map shows the difference in the coefficient of fitting. (C)There is an upward trend of the genes related to the neurodegenerative pathway in the inner retinal layer at the early stage. The fitting coefficients for the INL layers of the early stage were greater than 1. In other retinal layers of the early stage and in all the retinal layers of the late stage (D), there is a downward trend of the neurodegenerative pathway in the rd1 group compared to the control group.
Comment 16:
Line 398 “The present study is not limited to the retina, but also includes the RPE layer.”
The RPE is not referenced in any of the results or sampling technique.
Response to comment 16:
Thank you very much for your suggestion. Firstly, we apologize for any misunderstanding caused by our description. Secondly, we have removed the description related to RPE from the text. “The present study is not limited to the retina, but also includes the RPE layer. The findings suggested that the RPE layer plays an important role in the pathological changes of the late stage of RP. Both photoreceptor apoptosis in the early stage and neovascularization in the late stage are mainly located in the outer retina. The findings also suggested that vascular dysfunction mainly occur in close to the RPE layer. The RPE is essential for the normal physiological function of the outer layer of the retina, engulfing detached extracellular fragments of photoreceptors and secreting neurotrophic and vasotrophic types of growth factors [23]. Studies suggest that the disruption of the blood retinal barrier and structural changes in the choroidal vasculature may play an important role in the progression of advanced RP. The choroidal and RPE layers may provide important information to help in understanding the pathophysiology of advanced RP and finding potential treatment strategies for patients with RP [24,25]. It is not clear when the burden of oxidative stress exceeds the physiological signals of the RPE layer and becomes pathological. Furthermore, loss of photoreceptors precedes degeneration of the retinal vasculature, while RPE cells migrate towards the retinal vascular system after the loss of photoreceptors. It is unclear whether the genetic variants expressed in the RPE have a significant effect on the structure of the retinal vascular system in RP. The spatial transcriptomic based on micro-dissection was able to obtain samples not only from the retinal ganglion cell layer and photoreceptor cell layer, but also from the RPE layer. Precise sampling with spatial location information contained allows for accurate sampling of the RPE layer.” Originally, the reason for mentioning the RPE layer was because it was possible to collect a portion of the RPE layer when sampling the outer retina near the RPE layer. To avoid misunderstandings in reading, we have deleted this section. Finally, thank you again for your valuable suggestion, which made our article easier to understand.
Comment 17:
Line 478-480 “Moreover, the spatial transcriptomics is not single-cell and contains about ten cells per sampling spot, which does not distinguish well between the effects of cell types of RP.”
Authors should also acknowledge that Smart-seq(2) only captures mRNA.
Response to comment 17:
Thank you very much for your suggestion. Indeed, one of our drawbacks is using Smart seq (2) only captures mRNA. We have added the following description in the main text. “Besides, Smart-seq2 only captures mRNA.” Thank you again for your valuable suggestion.
Thank you again for reading and for your corrections. We hope that the correction will meet with approval.
At last, all of us appreciate for your warm work earnestly. Thank you very much for your comments and suggestions.
Authors’ contributions
Author contributions: Ying Zhou: investigation, formal analysis, writing original draft. Yuqi Sheng: data analysis. Min Pan: investigation, methodology. Jing Tu: supervision, resources. Xiangwei Zhao: supervision, resources. Qinyu Ge: methodology, conceptualization, writing review and editing. Zuhong Lu: supervision.
With best regards,
Yours sincerely,
Qinyu Ge
Mail address: geqinyu@seu.edu.cn

Reviewer 2 Report
This is a good paper, I have just a minor suggestion to the authors.
In my opinion the paper lack in some IHC staining i the main manuscript at the analysis time point selected as a double confirmation of their results.
Please add into the main manuscript the immunofluorescence confirmation and selected a better retinal sections as some of the presented are difficult to see, e.g. Fig. S7 panel D,I and E; Fig.S8 panel C and D; Fig. S9 P21-28 and 35. Please add also the letter to the panel of the figure S9 (A,B,C,D).
Author Response
Dear Editor and Reviewers:
First of all, thank you for your previous comments concerning our manuscript entitled “Spatial transcriptomic reveals the changes of early and late stages of rd1 model mice with retinitis pigmentosa”. Those comments are all valuable and very helpful for revising and improving our paper, as well as the important guiding significance to our research. We have carefully studied these comments and corrected the paper in the hope that it will be more suitable for International Journal of Molecular Sciences.
Reviewer 2:
This is a good paper, I have just a minor suggestion to the authors.
In my opinion the paper lack in some IHC staining i the main manuscript at the analysis time point selected as a double confirmation of their results.
Please add into the main manuscript the immunofluorescence confirmation and selected a better retinal sections as some of the presented are difficult to see, e.g. Fig. S7 panel D,I and E; Fig.S8 panel C and D; Fig. S9 P21-28 and 35. Please add also the letter to the panel of the figure S9 (A,B,C,D).
Response:
Thank you for reviewing and commenting on the manuscript. It is our honor to receive your approval of the manuscript. We have placed the attached IHC chart in the body of the text. Besides, we have replaced the unclear images in the picture. The final diagram is shown below.
Figure 4. Immunofluorescence validation of the Vascular endothelial growth factor (VEGF) pathway. (A)In control mice, immunoreactivity of EFNB2 was located throughout the retinal layers and remained generally unchanged across ages. (See Figure S10 for other time points) (B) The expression results of EFNB2 in rd1 mice at day P14. (C) The expression results of EFNB2 in rd1 mice at day P21. (D) The expression results of Efnb2 in rd1 mice at day P28. (E) The expression results of Efnb2 in rd1 mice at day P35.
Figure S9. Immunofluorescence expression of EFNB2 protein in normal C57BL/6J mice.
(A) Immunofluorescence expression of EFNB2 protein of C57BL/6J mice in P14.
(B) Immunofluorescence expression of EFNB2 protein of C57BL/6J mice in P21.
(C) Immunofluorescence expression of EFNB2 protein of C57BL/6J mice in P28.
(D) Immunofluorescence expression of EFNB2 protein of C57BL/6J mice in P35.
Immunofluorescence expression of EFNB2 protein in normal C57BL/6J mice did not differ significantly from P14 to P35.
Thank you again for reading and for your corrections. We hope that the correction will meet with approval.
At last, all of us appreciate for your warm work earnestly. Thank you very much for your comments and suggestions.
Authors’ contributions
Author contributions: Ying Zhou: investigation, formal analysis, writing original draft. Yuqi Sheng: data analysis. Min Pan: investigation, methodology. Jing Tu: supervision, resources. Xiangwei Zhao: supervision, resources. Qinyu Ge: methodology, conceptualization, writing review and editing. Zuhong Lu: supervision.
With best regards,
Yours sincerely,
Qinyu Ge
Mail address: geqinyu@seu.edu.cn
